# A Reinforcement Learning Approach to Estimating Long-term Effects in Nonstationary Environments

## Abstract

Randomized experiments (a.k.a. A/B tests) are a powerful tool for estimating treatment effects, to inform decisions making in business, healthcare and other applications. In many problems, the treatment has a lasting effect that evolves over time. A limitation with randomized experiments is that they do not easily extend to measure long-term effects, since running long experiments is time-consuming and expensive. In this paper, we take a reinforcement learning (RL) approach that estimates the average reward in a Markov process. Motivated by real-world scenarios where the observed state transition is nonstationary, we develop a new algorithm for a class of nonstationary problems, and demonstrate promising results in two synthetic datasets and one online store dataset.

## 1 Introduction

Randomized experiments (a.k.a. A/B tests) are a powerful tool for estimating treatment effects, to inform decisions making in business, healthcare and other applications. In an experiment, units like customers or patients are randomly split into a treatment bucket and a control bucket. For example, in a rideshare app, drivers in the control and treatment buckets are matched to customers in different ways (e.g., with different spatial ranges or different ranking functions). After we expose customers to one of these options for a period of time, usually a few days or weeks, we can record the corresponding customer engagements, and run a statistical hypothesis test on the engagement data to detect if there is a statistically significant difference in customer preference of treatment over control. The result will inform whether the app should launch the treatment or control.

While this method has been widely successful (e.g., in online applications (Kohavi et al., 2020)), it typically measures treatment effect *during* the short experiment window. However, in many problems, a treatment has a lasting effect that evolves over time. For example, a treatment that increases installation of a mobile app may result in a drop of short-term profit due to promotional benefits like discounts. But the installation allows the customer to benefit from the app, which will increase future engagements and profit in the long term. A limitation with standard randomized experiments is that they do not easily extend to measure long-term effects. We can run a long experiment for months or years to measure the long-term impacts, which however is time-consuming and expensive. We can also design proxy signals that are believed to correlate with long-term engagements (Kohavi et al., 2009), but finding a reliable proxy is challenging in practice. Another solution is the surrogacy method that estimates *delayed* treatment impacts from surrogate changes during the experiment (Athey et al., 2019). However, it does not estimate long-term impacts resulting from long-term treatment exposure, but rather from short-term exposure during the experiment.

Shi et al. (2022b) mitigates the limitation of standard randomized experiment by framing the long-term effect as a reinforcement learning (RL) problem. Their method is closely related to recent advances in infinite-horizon off-policy evaluation (OPE) (Liu et al., 2018; Nachum et al., 2019a; Xie et al., 2019; Kallus & Uehara, 2020; Uehara et al., 2020; Chandak et al., 2021). However, their solution relies on stationary Markov assumption, which fails to capture the real-world nonstationary dynamics. Motivated by real-world scenarios where the observed state transitions are nonstationary, we consider a class of nonstationary problems, where the observation consists of two additive terms: an endogenous term that follows a stationary Markov process, and an exogenous

term that is time-varying but independent of the policy. Based on this assumption, we develop a new algorithm to jointly estimate long-term reward and the exogenous variables.

Our contributions are threefold. First, it is a novel application of RL to estimate long-term treatment effects, which is challenging for standard randomized experiments. Second, we develop an estimator for a class of nonstationary problems that are motivated by real-world scenarios, and give a preliminary theoretical analysis. Third, we demonstrate promising results in two synthetic datasets and one online store dataset.

## 2 BACKGROUND

### 2.1 LONG-TERM TREATMENT EFFECTS

Let $\pi_0$ and $\pi_1$ be the control and treatment policies, used to serve individual in respective buckets. In the rideshare example, a policy may decide how to match a driver to a nearby request. During the experiment, each individual (the driver) is randomly assigned to one of the policy groups, and we observe a sequence of behavior features of that individual under the influence of the assigned policy. We use variable $D \in \{0, 1\}$ to denote the random assignment of an individual to one of the policies. The observed features are denoted as a sequence of random variable in $\mathbb{R}^d$

$$O_0, O_1, \ldots, O_t, \ldots,$$

where the subscript $t$ indicates time step in the sequence. A time step may be one day or one week, depending on the application. Feature $O_t$ consists of information like number of pickup orders. We are interested in estimating the difference in average long-term reward between treatment and control policies:

$$\Delta = \mathbb{E}[\sum_{t=0}^{\infty} \gamma^t R_t | D = 1] - \mathbb{E}[\sum_{t=0}^{\infty} \gamma^t R_t | D = 0], \tag{1}$$

where $\mathbb{E}$ averages over individuals and their stochastic sequence of engagements, $R_t = r(O_t)$ is the reward signal (e.g., customer rating) at time step $t$, following a pre-defined reward function $r : \mathbb{R}^d \to \mathbb{R}$, and $\gamma \in (0, 1)$ is the discounted factor. The discounted factor $\gamma$ is a hyper-parameter specified by the decision maker to indicate how much they value future reward over the present. The closer $\gamma$ is to 1, the greater weight future rewards carry in the discounted sum.

Suppose we have run a randomized experiment with the two policies for a short period of $T$ steps. In the experiment, a set of $n$ individuals are randomly split and exposed to one of the two policies $\pi_0$ and $\pi_1$. We denote by $d_j \in \{0, 1\}$ the policy assignment of individual $j$, and $\mathcal{I}_i$ the index set of individuals assigned to $\pi_i$, i.e., $j \in \mathcal{I}_i$ iff $d_j = i$. The in-experiment trajectory of individual $j$ is:

$$\tau_j = \{o_{j,0}, o_{j,1}, \ldots, o_{j,T}\}.$$

The in-experiment dataset is the collection of all individual data as $\mathcal{D}_n = \{(\tau_j, d_j)\}_{j=1}^n$. Our goal is to find an estimator $\hat{\Delta}(\mathcal{D}_n) \approx \Delta$.

### 2.2 ESTIMATION UNDER STATIONARY MARKOVIAN DYNAMICS

Inspired by recent advances in off-policy evaluation (OPE) (e.g. Liu et al., 2018; Nachum et al., 2019b), the simplest assumption is a fully observed Markov Process that the observation in each time step can fully predict the future distribution under a stationary dynamic kernel. In this paper, we assume the dynamics kernel and reward function are both linear, following the setting in Parr et al. (2008). Linear representations are popular in the RL literature (e.g., Shi et al., 2022b) , and often preferable in industrial applications due to simplicity and greater model interpretability.

**Assumption 2.1.** *(Linear Dynamics) there is a matrix $M_i$ such that*

$$\mathbb{E}[O_{t+1}|O_t = o, D = i] = M_i o, \ \forall t \in \mathbb{N}, i \in \{0, 1\}. \tag{2}$$

**Remark 2.2.** *Unlike standard RL, we don't have an explicit action for a policy. The difference between the control and treatment policy is revealed by different transition matrix $M$.*

**Assumption 2.3.** *(Linear Reward) There is a coefficient vector $\theta_r \in \mathbb{R}^d$ such that*

$$r(O_t) = \theta_r^\top O_t, \ \forall t \in \mathbb{N}. \tag{3}$$

**Remark 2.4.** *The reward signal may be one of the observed features. For example, if we are interested in customer rating, and rating is one of the observe features, then $\theta_r$ is just a one-hot vector with $1$ in the corresponding coordinate. When the reward is complex with unknown coefficient, we can use ordinary least-squares to estimate the coefficient $\theta_r$.*

**Proposition 2.5.** *Under Assumption 2.1 and 2.3, if the spectral norm of $M_i$ is smaller than $\frac{1}{\gamma}$, then the expected long-term reward of policy $\pi_i$, $v(\pi_i) := \mathbb{E}[\sum_{t=0}^{\infty} \gamma^t R_t | D = i]$, can be obtained by:*

$$v(\pi_i) = \theta_r^\top (I - \gamma M_i)^{-1} \bar{O}_0^{(i)}, \quad where \ \ \bar{O}_0^{(i)} := \mathbb{E}[O_0 | D = i]. \tag{4}$$

The only remaining step is to estimate $\bar{O}_0^{(i)}$ and $M_i$. The former can be directly estimated from the Monte Carlo average of the experimental data: $\hat{O}_0^{(i)} = \frac{1}{n_i} \sum_{j \in \mathcal{I}_i} o_{0,j}$, where $n_i = |\mathcal{I}_i|$ is the number of individuals assigned to policy $\pi_i$. To estimate the latter, we may use ordinary least-squares on observed transitions:

$$\hat{M}_i = \left( \sum_{j \in \mathcal{I}_i} \sum_{t=0}^{T-1} o_{j,t+1} o_{j,t}^\top \right) \left( \sum_{j \in \mathcal{I}_i} \sum_{t=0}^{T-1} o_{j,t} o_{j,t}^\top \right)^{-1}. \tag{5}$$

The detailed derivation can be found in (Parr et al., 2008). Once we get the estimated value of $\hat{v}_i \approx v(\pi_i)$, the long term impact in Eq. (1) can be estimated as:

$$\hat{\Delta} = \hat{v}_1 - \hat{v}_0.$$

**Remark 2.6.** *Although this a model-based estimator, it is equivalent to other OPE estimator in general under linear Markovian assumption (e.g., Nachum et al., 2019b; Duan et al., 2020; Miyaguchi, 2021) and it enjoys similar statistical guarantees as other OPE estimators.*

## 3 OUR METHOD

In Section 2.2, we assumed the observation $O_t$ follows a stationary Markov process, and derived a model-based closed-form solution based on linear reward Assumption 2.3.

In reality, this model assumption has two major limitations. First, real-world environments are nonstationary. For example, in a hotel reservation system, seasonality heavily influences the prediction of the future booking count. Our stationary assumption does not capture those seasonal changes, resulting in poorly learned models and inaccurate predictions of long-term treatment effects. Second, in practice, we are unable to ensure that observed features fully capture the dynamics. OPE methods based on stationary and full observability assumptions are unlikely to work robustly in complex, real-life scenarios.

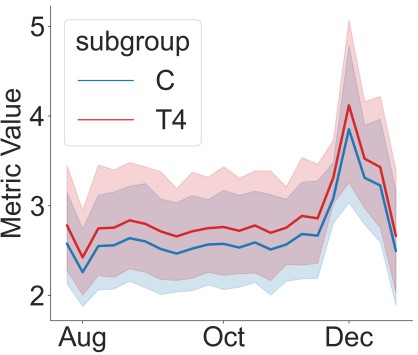

Figure 1 illustrates nonstationarity in data from an online store (see Section 5 for more details). The figure shows how the weekly average of a business metric changes in a span of 5 months, for two policies (C for control, and T4 for treatment). Such highly non-stationary data, especially during special seasons towards the right end of the plot, are common.

Figure 1: An example of non-stationarity. The weekly average metric value is highly non-stationary during holiday season.

However, the difference of the two policy groups remains much more stable. This is expected as both policies are affected by the same exogenous affects (seasonal variations in this example).

Figure 1 motivates a relaxed model assumption (Section 3.1), by introducing a non-stationary exogenous component on top of a stationary hidden state $S_t$. Our new assumption is that the observation $O_t$ can be decomposed additively into two parts: an endogenous part still follows a stationary Markovian dynamic for each policy group (treatment or control); and an exogenous part which is time-varying and shared across all groups. Based on the new assumption we propose an alternating minimization algorithm that jointly estimates both transition dynamics and exogenous variables.

### 3.1 Nonstationary Model Relaxation

We assume there is an exogenous noise vector $z_t$ for each time step $t$, to represent the linear additive exogenous noise in the uncontrollable outside world such as seasonal effect, which applies uniformly to every individual under each treatment bucket. We relax Assumption 2.1 as the following:

**Assumption 3.1.** *(Linear Additive Exogenous Noise) the observational feature $O_t$ is the sum of the endogenous hidden features and the time-varying exogenous noise $z_t$.*

$$O_t = S_t + z_t, \ \forall t \in \mathbb{N}.$$

*where $z_t$ does not depend on policy or any individual in the experiments and $S_t$ follows the linear Markovian kernel with transition matrix $M_i$:*

$$\mathbb{E}[S_{t+1}|S_t = s, D = i] = M_i s, \ \forall t \in \mathbb{N}, i \in \{0, 1\}. \tag{6}$$

**Remark 3.2** (Explanation of the Linear Additive Model). *Our linear additive model is inspired by the parallel trend assumption in the Difference-in-Difference (DID) estimator (Lechner et al., 2011). In real-world environments, it is impossible to capture all the covariates that may effect the dynamics. The linear additive exogenous noise $z_t$ can be seen as the drive from the outside that is both unobserved and uncontrol. For example, in an intelligent agriculture system, the highly non-stationary weather condition can be seen as exogenous which we cannot control, but the amount of water and fertilizer that affect the growth of the plant can be seen as the hidden state that is controlled by a pre-defined stationary policy. And we add up those two factors as the features (e.g., the condition of the crop) we observed in the real world.*

From Assumption 3.1 and linear reward function assumption in 2.3, the closed form of $v(\pi_i)$ can be rewritten as:

**Proposition 3.3.** *Under Assumption 3.1 and 2.3, and suppose $v(z_\infty) := \sum_{t=0}^{\infty} \gamma^t z_t < \infty$. Suppose the spectral norm of $M_i$ is smaller than $\frac{1}{\gamma}$, the expected long-term reward can be obtained by:*

$$v(\pi_i) = \theta_r^\top (I - \gamma M_i)^{-1} \bar{S}_0^{(i)} + v(z_\infty), \ \text{where} \ \bar{S}_0^{(i)} = \mathbb{E}[S_0|D = i]. \tag{7}$$

The long-term reward in Eq. (7) contains $v(z_\infty)$, which depends on the unknown exogenous noise sequence outside of the experimental window and thus is unpredictable. However, the long term treatment effect, $\Delta(\pi_1, \pi_0) = v(\pi_1) - v(\pi_0)$, cancels out the dependency on that exogenous term $v(z_\infty)$. For simplicity, we redefine $v(\pi_i) = \theta_r^\top (I - \gamma M_i)^{-1} \bar{S}_0^{(i)}$ without the term of $v(z_\infty)$. Therefore, the only thing we need to estimate is $\bar{S}_0^{(i)}$ and $M_i$. Once we have the access of $z_0$, we can estimate $\bar{S}_0^{(i)}$ similarly as Monte Carlo sample: $\hat{S}_0 = \frac{1}{n_i} \sum_{j \in \mathcal{I}_i} o_{0,j} - \hat{z}_0$. The next question is how to estimate in-experiment exogenous variable $z_t$ and the underlying transition kernels.

### 3.2 Optimization Framework

We propose to optimize $\{z_t\}_{1 \le t \le T}$ and $\{M_0, M_1\}$ jointly under a single loss function, with the same spirit of reducing the reconstruction loss of each transition pair similar to the model-based approach.

For each individual $j$ in treatment group $i$, Assumption 3.1 implies that at time step $t + 1$, the observation $o_{j,t+1}$ can be written as:

$$o_{j,t+1} - z_{t+1} = M_i(o_{j,t} - z_t) + \varepsilon_{j,t}, \ \forall j \in \mathcal{I}_i, \ 1 \le t \le T - 1, \tag{8}$$

where $\varepsilon_{j,t}$ is a noise term with zero mean, so that $M_i(o_{j,t} - z_t) = \mathbb{E}[S_{t+1}|S_t = o_{j,t} - z_t, D = i]$.

Inspired by Eq. (8), given observation history $\mathcal{D}_n$, in order to minimize the empirical reconstruct risk by each transition pair $(o_{j,t}, o_{j,t+1})$, we construct the following loss function

$$\mathcal{L}(M_0, M_1, \{z_t\}_{1 \le t \le T}; \mathcal{D}_n) = \sum_{i=0}^{1} \sum_{j \in \mathcal{I}_i} \sum_{t=0}^{T-1} \|o_{j,t+1} - z_{t+1} - M_i(o_{j,t} - z_t)\|_2^2. \tag{9}$$

To simplify the notation, Eq. (9) can be rewritten as a vectorized form

$$\mathcal{L}(M_0, M_1, z; \mathcal{D}_n) = \sum_{i=0}^{1} \sum_{j \in \mathcal{I}_i} \|A_i(o_j - z)\|_2^2, \tag{10}$$

---

**Algorithm 1** Estimating Long-Term Effect Under Non-stationary Dynamics

---

**Input**: In-experiment training Data $\mathcal{D}_n = \{(\tau_j, d_j)\}_{j=1}^n$, where $\tau_i = (o_{j,0}, o_{j,1}, \ldots, o_{j,T})$ is the in-experiment observation features for individual $j$, $d_j \in \{0, 1\}$ is the indicator of which policy group individual $j$ is assigned to.

**Initialize** the estimation of exogenous noise $\hat{z} = 0$.

**Optimization**:

**while** not convergent **do**

    Update $M_i$ as the ordinary least square solution given the current $\hat{z}$:

$$\hat{M}_i = \left( \sum_{j \in \mathcal{I}_i} \sum_{t=1}^{T-1} (o_{j,t+1} - \hat{z}_{t+1})(o_{j,t} - \hat{z}_t)^\top \right) \left( \sum_{j \in \mathcal{I}_i} \sum_{t=1}^{T-1} (o_{j,t} - \hat{z}_t)(o_{j,t} - \hat{z}_t)^\top \right)^{-1}.$$

    Update $\hat{z}$ according to Eq. (12):

$$\hat{z} = (n_0 G_0 + n_1 G_1)^{-1} (\sum_{i=0}^{1} \sum_{j \in \mathcal{I}_i} G_i o_j).$$

**end while**

**Evaluation**:

    Compute $\hat{v}_i = \theta_r^\top (I - \gamma \hat{M}_i)^{-1} \left( \hat{O}_0^{(i)} - \hat{z}_0 \right)$, where $\hat{O}_0^{(i)} = \frac{1}{n_i} \sum_{j \in \mathcal{I}_i} o_{0,j}$.

    Output the long-term impact estimation as $\hat{\Delta} = \hat{v}_1 - \hat{v}_0$.

---

where $o_j = \begin{pmatrix} o_{j,0} \\ o_{j,1} \\ \ldots \\ o_{j,T} \end{pmatrix}$, and $z = \begin{pmatrix} z_0 \\ z_1 \\ \ldots \\ z_T \end{pmatrix}$ are column vector aggregate over the experiment time horizon, and $A_i$ is a $dT \times d(T+1)$ matrix constructing by a block matrix $M_i$:

$$A_i = \begin{bmatrix} -M_i & I & \ldots & & \\ 0 & -M_i & \ldots & & \\ & & \ldots & & \\ & & \ldots & I & 0 \\ & & \ldots & -M_i & I \end{bmatrix}_{dT \times d(T+1)}. \tag{11}$$

### 3.3 ALTERNATING MINIMIZATION

To reconstruct $M_i$ and $z$, we apply alternating minimization on the loss function $\mathcal{L}(M_0, M_1, z; \mathcal{D}_n)$ in Eq. (10). By looking at the zero-gradient point of the loss function, under proper non-degenerate assumption (see Appendix for details), we have:

**Proposition 3.4.** *Suppose $(n_0 G_0 + n_1 G_1)$ is nonsingular, the minimizer of $z$ given $M_i$ is a closed-form solution in the followings:*

$$\arg \min_z \mathcal{L}(M_0, M_1, z; \mathcal{D}_n) = (n_0 G_0 + n_1 G_1)^{-1} (\sum_{i=0}^{1} \sum_{j \in \mathcal{I}_i} G_i o_j), \quad \text{where } G_i = A_i^\top A_i. \tag{12}$$

The minimizer of $M_i$ given $z$ is similar to Eq. (5), except that we subtract the exogenous part $z_t$ from the observation:

$$\arg \min_{M_0, M_1} \mathcal{L}(M_0, M_1, z; \mathcal{D}_n)$$

$$:= \left( \sum_{j \in \mathcal{I}_i} \sum_{t=1}^{T-1} (o_{j,t+1} - \hat{z}_{t+1})(o_{j,t} - \hat{z}_t)^\top \right) \left( \sum_{j \in \mathcal{I}_i} \sum_{t=1}^{T-1} (o_{j,t} - \hat{z}_t)(o_{j,t} - \hat{z}_t)^\top \right)^{-1}. \tag{13}$$

The final optimization process is summarized in Algorithm 1.

### 3.4 THEORETICAL ANALYSIS

We give a preliminary theoretical analysis in this section to give readers some insights on how good our estimator is once a partial oracle information is given. We will extend our analysis to quantify the error of the estimator at the convergence state of alternating minimization in future work.

To simplify our analysis, we first assume we get access to the true transition matrix $M_i$, and our goal is to quantify the error between $\hat{v}(\pi_i)$ and the true policy value $v(\pi_i)$ for each policy $\pi_i$.

**Proposition 3.5.** *Suppose we have bounded noise and matrices under Assumption A.1 and Assumption A.2, and suppose $n_0 = n_1 = \frac{n}{2}$ is equally divided. When we get access of the oracle transition matrix $M_i = M_i^*$, $i \in \{0,1\}$, let $\hat{z} = \arg\min_z \mathcal{L}(M_0^*, M_1^*, z; \mathcal{D}_n)$. If we plugin $\hat{z}$ in the estimation of $\hat{v}(\pi_i)$, we will have*

$$|\hat{v}(\pi_i) - v(\pi_i)| = \mathcal{O}(\frac{1}{\sqrt{n}}),$$

*with probability at least $1 - \delta$.*

In the second analysis we assume that we get an accurate $z$. In this case, the estimation of $\hat{M}$ reduces to the stationary assumption case in Assumption 2.1 where the hidden state variable $s_t = o_t - z_t$ is fully recovered. We follow the analysis (e.g., Duan et al., 2020; Miyaguchi, 2021) of linear MDP to characterize the error.

**Proposition 3.6** (Proposition 11 in Miyaguchi (2021)). *Suppose we get access to the oracle exogenous noise $z$ during the experimental period, let $\hat{M}_i = \arg\min_{M_i} \mathcal{L}(\{M_i\}, z^*; \mathcal{D}_n)$ in Eq. (13). Under the assumption in Proposition 11 in Miyaguchi (2021), with the plugin estimator $\hat{v}$ with $\hat{M}_i$, we have:*

$$|\hat{v}(\pi_i) - v(\pi_i)| = \mathcal{O}(n^{-\frac{1}{2d+2}}),$$

*with probability at least $1 - \delta$.*

### 3.5 PRACTICAL CONSIDERATIONS

**Regularize the Transition Dynamic Matrices.** Degenerated case may happen during the alternating minimization when either 1) the spectral norm is too large, i.e. $\|M_i\|_2 \geq \frac{1}{\gamma}$, leading the long-term operator $(I - \gamma M_i)^{-1} = \sum_{t=0}^{\infty} \gamma^t M_i^t$ diverges in Eq. (7), or 2) the matrix inversion calculation of $M_i$ in Eq. (13) is not well-defined. To avoid those scenarios and stabilize the computation procedure, we add a regularization term of $M_i$ as $\lambda_i \|M_i - I_d\|_2^2$ in our experiment. The intuition is that the transition matrix should be close to identity matrix as in practice the treatment policy typically deviates from the control policy in an incremental manner.

After adding the regularization, the closed-form minimizer of $M_i$ of the regularized loss function becomes:

$$M_i = \left(\lambda_i I_d + \sum_{j \in \mathcal{I}_i} \sum_{t=1}^{T-1} (o_{j,t+1} - z_{t+1})(o_{j,t} - z_t)^\top\right)\left(\lambda_i I_d + \sum_{j \in \mathcal{I}_i} \sum_{t=1}^{T-1} (o_{j,t} - z_t)(o_{j,t} - z_t)^\top\right)^{-1}.$$

**Regularize the Exogenous Variable.** There is a challenge in deriving the closed-form $z$ in Eq. (12) where $n_0 G_0 + n_1 G_1$ can be degenerated or nearly degenerated. By definition, $G_i$ is always singular. Moreover, if there is no control of the minimal eigenvalue of $(n_0 G_0 + n_1 G_1)$, e.g. close to zero, the update step on $z$ is uncontrolled and the variance of noise can be magnified in the direction of the minimal eigenvector. Therefore it is crucial to regularize $z$.

To tackle the possible degenerated circumstances, one natural idea is to include regularization of the $\ell_2$ norm of $z$, where the regularized loss function can be written as:

$$\mathcal{L}_\lambda(z, M_0, M_1; \mathcal{D}) = \mathcal{L}(z, M_0, M_1; \mathcal{D}) + \lambda_z \|z\|_2^2. \tag{14}$$

Its corresponding minimizer of $\hat{z}$ can be written as:

$$\hat{z} = (\lambda I + n_0 G_0 + n_1 G_1)^{-1}(\sum_{i=0}^{1} \sum_{j \in \mathcal{I}_i} G_i o_j),$$

where $I$ is the identity matrix of dimension $d \times (T + 1)$. It is worth mentioning that when the regularization parameter $\lambda$ increases to infinity, $z$ will go to 0, and the solution reduces to the stationary case in Assumption 2.1.

**Extend to Multiple Treatment Policies** The optimization framework can be easily extend to multiple treatment policies case. Suppose we have $k$ different treatment policies $\pi_1, \pi_2, \cdots, \pi_k$ and let $\pi_0$ be the control policy, the closed form solution for $\hat{z}$ under multiple dataset of different treatment groups can be derived as

$$\hat{z}_\lambda = (\lambda I + \sum_{i=0}^{k} n_i G_i)^{-1} (\sum_{i=0}^{k} \sum_{j \in \mathcal{I}_i} G_i o_j).$$

And the closed-form update for $M_i$ stays the same. The final estimation of the treatment effect for policy $\pi_i$ is $\hat{\Delta} = \hat{v}_i - \hat{v}_0$.

## 4 RELATED WORK

**Estimating long-term treatment effects** Our work is related to causal inference with temporal data. The surrogate index method (Athey et al., 2019; 2020) makes a different assumption that the long-term effect is independent of the treatment conditioned on the surrogate index measured during the experiment. It then estimates long-term impacts resulting from *short-term* exposure during the experiment. In contrast, our work aims to estimate long-term impacts resulting from *long-term* exposure. Time series methods (e.g. Bojinov & Shephard, 2019) require probabilistic treatments, which allow an individual to be exposed to different treatments at different time periods during an experiment. They then estimate the temporal treatment effect, which is averaged over all the temporal steps, differs from traditional treatment effect which is averaged over randomized individuals.

Our method draws inspirations from off-policy evaluation(OPE) and related areas, whose goal is to estimate the long-term policy value, usually from a offline dataset collected under different policies. Most early work focuses on the family of inverse propensity score estimators that are prone to high variance in long-horizon problems (e.g., Precup et al., 2000; Murphy et al., 2001; Jiang & Li, 2016). Recently, there are growing interests in long- and even infinite-horizon settings (Liu et al., 2018; Nachum et al., 2019a; Xie et al., 2019; Tang et al., 2020; Uehara et al., 2020; Dai et al., 2020; Chandak et al., 2021). In particular, Shi et al. (2022b) considers a similar problem of estimating long-term impacts, which is comparable to our stationary baseline. However, these methods either rely on the stationarity assumption that is violated in many applications, or consider the general nonstationary Markov decision process (Kallus & Uehara, 2020) that does not leverage domain-specific assumptions.

**RL in nonstationary or confounded environments** Our model is a special case of Partially Observable Markov Decision Process (POMDP) (Åström, 1965; Kaelbling et al., 1998). OPE in general POMDPs remains challenging, unless various assumptions are made (e.g., Tennenholtz et al., 2020; Bennett et al., 2021; Shi et al., 2022a). Most assumptions are on the causal relation of the logged data, such as relation between state, action and confounded variable. In contrast, we make an assumption motivated by real-world data, which allows our estimator to cancel out exogenous variables from observations.

Our assumption is also related to MDP with Exogenous Variables (e.g., Dietterich et al., 2018; Chitnis & Lozano-Pérez, 2020), and Dynamics Parameter MDP (DPMDP) or Hidden Paramter MDP (HiP-MDP) (Al-Shedivat et al., 2017; Xie et al., 2020). For exogenous variable, they assume observation features can be partitioned into two groups, where the exogenous group is not affected by the action and the endogenous group evolve as in a typical MDP. The major challenge is infer the right partition. Several recent works (e.g Misra et al., 2020; Du et al., 2019; Efroni et al., 2021) combine exogenous variable with rich observation in RL. This is different from our assumption where we assume the observation is a *sum* of both parts, which is a more natural assumption in applications like e-commerce. For DPMDP and Hip-MDP, they assume a meta task variable which is non-stationary and changed across time but the task variable dynamic can be captured by a sequential model. Our

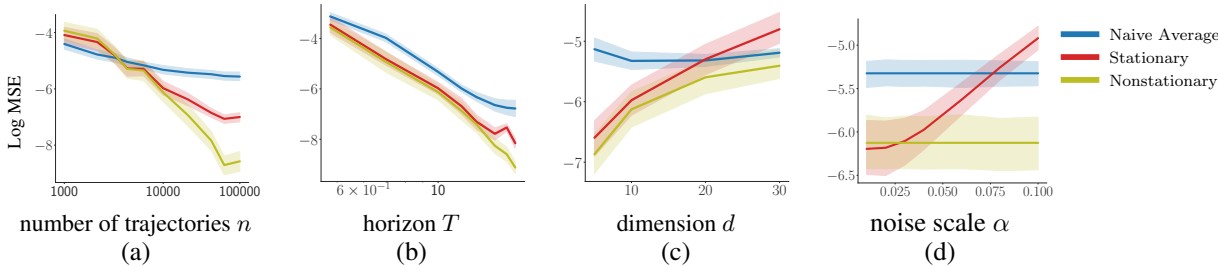

Figure 2: Results of Synthetic Environment. We vary parameters in the simulation to compare the logarithmic MSE of various estimators: (a) number of trajectories; (b) horizon; (c) observation feature dimension; (d) scale of the exogeneous noise.

assumption can be viewed as a linear special case but our focus is not to better characterize the system but is to remove the exogenous part for better predictions.

## 5 EXPERIMENTS

We evaluate our methods in three problems: a synthetic dataset, a dataset from the Type-1 Diabete RL simulator (Xie, 2019), and a real-world dataset from an online store. The ground truth $\Delta$ is computed either from a true simulator or using the average of the real experimental data under a long time period. We compare our methods based on plug-in estimator of the stationary solution in Eq. (4), its non-stationary variant in Algorithm 1, and an *Naive Average* baseline. The baseline directly uses the short-term reward average as the estimate of the long-term effect.

### 5.1 SYNTHETIC SIMULATION

The synthetic environment generates 4 randomized matrix $M_i$ for policies $\{\pi_i\}_{i=0}^3$ and a trajectory of randomized exogenous noise $\{z_t\}_{t=0}^T$. See details of the synthetic dynamic in Appendix C. The randomized sequence follows the non-stationary dynamics with a parameter $\alpha$ controlling the scale of the exogenous noise: $o_{j,t} = s_{j,t} + \alpha z_t, \ \forall j, t$. We collect $n$ trajectories for each policy until $t = T$ (w/ varying $T$). We vary the parameters of the generating sequences: number $n$ of trajectories, horizon $T$, data dimension $d$, and scale $\alpha$ of the exogenous noise. We plot the logarithmic Mean Square Error (MSE) for each method in Figure 2. The result shows that our estimator method (the green line) clearly outperforms all other baselines. Moreover, Figure 2(d) shows the increase of the scale of the exogenous noise does not affect estimation accuracy of our method.

### 5.2 TYPE-1 DIABETE SIMULATOR

This environment is modified based on an open-source implementation[1] of the FDA approved Type-1 Diabetes simulator (T1DMS) (Man et al., 2014). The environment simulates two-day behavior in an in-silico patient's life. Consumption of a meal increases the blood-glucose level in the body. If the level is too high, the patient suffers from hyperglycemia. If the level is too low, the patient suffers from hypoglycemia. The goal is to control the blood glucose level by regulating the insulin dosage to minimize the risk associated with both hyperglycemia and hypoglycemia.

We modify the Bagal and Bolus (BB) policy (Bastani, 2014) (control policy) in the codebase and set two glucose target levels and different noise levels as our two treatment policies. We collect information in the first 12-hour of all the three policies with 5000 randomized patients in each policy group and use those information to predict the long-term effect. The observation feature is 2-dimensional: glucose level (CGM) and the amount of insulin injection. The non-stationarity comes from the time and amount of the consumption of a meal, which is time varying, but otherwise shared by all patients. We average a 2-day simulation window over random $250,000$ patients as ground truth treatment effect between policy groups.

---

[1]https://github.com/jxx123/simglucose

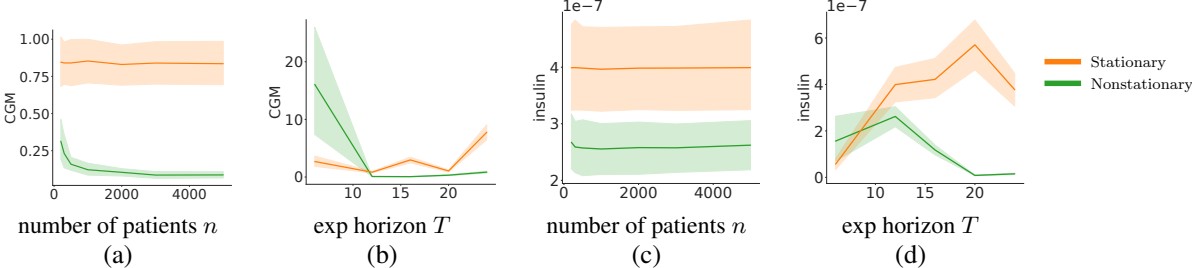

Figure 3: Results of Type-1 Diabete Environment. We vary two parameters in the simulation, the number of patients and the in-experiment horizon, to compare the performance for different methods under two different evaluation metrics.

Similar to the synthetic simulator, we vary the number of patients and the experimental period. Figure 3 shows that the non-stationary method performs better in the prediction accuracy compared to stationary method in both predictions of CGM and the amount of insulin injection. Even though the simulator is non-linear, our simple linear additive exogenous noise assumption still captures the small local changes well, which is approximately linear.

## 5.3 DATA FROM AN ONLINE STORE

|  | Metric 1 | Metric 2 | Metric 3 | Metric 4 |
|---|---|---|---|---|
| Naive Average | 133.61% | 66.09% | 106.85% | **37.99**% |
| Stationary | 174.77% | **48.03**% | 98.31% | 110.58% |
| Non-stationary | **64.68**% | 62.00% | **48.04**% | 67.57% |

Table 1: Results in the online store dataset. The reported numbers are the median of MAPE over 7 different treatment policies. Columns correspond to business metrics of interest.

We test our methods under 4 long-running experiments in an online store with a total of 7 different treatment policies (some experiments have more than 1 treatment). Each experiment has 1 control policy. We evaluate 4 business metrics related to customer purchases in the store (Metrics 1-4), and use $d = 17$ features. All the experiments lasted for 12 weeks. We treat the first 5 weeks as the experiment window, and use data in those weeks to estimate long-term impacts of the 4 metrics. The trailing 7-week average of the metrics are used as ground true to evaluate accuracy of various estimators. Table 1 reports the median of the Mean Absolute Percentage Error (MAPE) of the estimators; See full results in Appendix C.

Given the high cost in such long-running experiments, we cannot collect more data points for comparison, and for computing statistical significance. That said, there is good evidence from the reported number that our method produces better predictions of long-term treatment effects than Naive Average. Furthermore, our method improves on the stationary baseline, suggesting the practical relevance of our nonstationary assumtion, and effectiveness of the proposed estimator.

## 6 CONCLUSIONS

In this paper we study how to estimate the long-term treatment effect by using only the in-experimental data in the non-stationary environment. We propose a novel non-stationary RL model and an algorithm to make prediction. A major limitation is the linear assumption in both the dynamics model and the additive exogenous part. Once the real world model includes a highly non-linear part, the prediction value can be biased. Future direction includes further relax our model to non-linear case to better capture the real world environment.

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

# Appendix

## A   PROOF

In this section, we provide detailed proof for the theorem in the main text, as a self-contained section, we briefly introduce the notation as below, and adopt the regularized, multiple policy groups settings in the appendix:

- $n$: number of total individuals.
- $\mathcal{I}_i$: the index set for policy $\pi_i$; $n_i = |\mathcal{I}_i|$ as the number of individual in under policy $\pi_i$.
- $k$ total number of different policy group.
- $\mathcal{D}_n$: dataset for $n$ individuals in the experimental period.

In the appendix, we denote the ground truth dynamic $M_i^*$ and the ground truth exogenous noise $z^*$ with a star $*$ to distinguish the variables $M_i$ and $z$ during optmization process.

### A.1   ASSUMPTIONS

The dynamic assumption of our linear additive exogenous noise assumption in Assumption 3.1 can be rewritten as the following equation:

$$M_i^*(o_{j,t} - z_t^*) = (o_{j,t+1} - z_{t+1}^*) + \varepsilon_{j,t}, \; \forall j \in \mathcal{I}_i, 0 \leq t \leq T - 1. \tag{15}$$

where $\varepsilon_{j,t}$ is a zero-mean noise. Let $\varepsilon_j = \begin{pmatrix} \varepsilon_{j,0} \\ \varepsilon_{j,1} \\ \cdots \\ \varepsilon_{j,T-1} \end{pmatrix} \in \mathbb{R}^{d \times T}$, $\{\varepsilon_j\}_{1 \leq j \leq n}$ forms a martingale:

$$\mathbb{E}[\varepsilon_j | \mathcal{F}_{j-1}] = 0, \tag{16}$$

where the filtration $\mathcal{F}_j = \{o_1, ..., o_{j-1}\}$ is the information up to the first $j - 1$ individuals.

We make addition bounded assumption on the zero-mean noise term for the proof:

**Assumption A.1** (Bounded Noise assumption). *Let $\varepsilon_j = M_i^*(o_{j,t} - z_t^*) - (o_{j,t+1} - z_{t+1}^*)$, $j \in \mathcal{I}_i$ be the residual of the transition under the true transition matrix $M_i^*$, we have*

$$\|\varepsilon_j\|_2 \leq C_\varepsilon, \; \forall j, \tag{17}$$

*where $C_\varepsilon$ is a uniform constant independent of policy assignment.*

For the empirical covariance matrix in the middle step of the calculation, we assume they are all bounded.

**Assumption A.2** (Bounded Norm for Matrices). *We make the following assumptions on matrices*

1. $\|M_i^*\| \leq C_{M_i} < \frac{1}{\gamma}, \; \forall i.$

2. $\|(\Lambda_n^*/n)^{-1}\| \leq C_\Lambda.$

### A.2   LOSS FUNCTION AND ALTERNATING MINIMIZATION

Our loss function can be written as:

$$\mathcal{L}(\{M_i\}_{1 \leq i \leq k}, z; \mathcal{D}_n) = \sum_{i=1}^{k} \sum_{j \in \mathcal{I}_i} \|A_i(z - o_j)\|_2^2 + \lambda_z \|z\|_2^2 + \sum_{i=1}^{k} \lambda_i \|M_i - I_d\|_F^2. \tag{18}$$

**Lemma A.3.** *Fix $\{M_i\}$, denote $G_i = A_i^\top A_i$ where $A_i$ is defined in Eq. (11), the minimization of $z = \arg\min_z \mathcal{L}(\{M_i\}_{1 \leq i \leq k}, z; \mathcal{D}_n)$ is*

$$z(\{M_i\}) = \left( \lambda_z I_{d \times (T+1)} + \sum_{i=1}^{k} n_i G_i \right)^{-1} \left( \sum_{i=1}^{k} \sum_{j \in \mathcal{I}_i} G_i o_j \right). \tag{19}$$

*Proof.* By taking the gradient of the loss function, we will have:

$$0 = \nabla_z \mathcal{L}(\{M_i\}_{1 \le i \le k}, z; \mathcal{D}_n)$$

$$= 2 \sum_{i=1}^{k} \sum_{j \in \mathcal{I}_i} G_i(z - o_j) + 2\lambda_z z$$

which implies

$$z(\{M_i\}) = \left( \lambda_z I_{d \times (T+1)} + \sum_{i=1}^{k} n_i G_i \right)^{-1} \left( \sum_{i=1}^{k} \sum_{j \in \mathcal{I}_i} G_i o_j \right).$$

Here, $G_i = A_i^\top A_i$ is semi-definite, so the inversion of the large matrix in the right side of the expression always exists. $\qquad\square$

Similarly we can get the minimizer of $M_i$ fixing $z$.

**Lemma A.4.** *By fixing $z$, the minimizer of $M_i$ can be written as*

$$M_i(z) = \left( \lambda_i I_d + \sum_{j \in \mathcal{I}_i} \sum_{t=1}^{T-1} (o_{j,t+1} - z_{t+1})(o_{j,t} - z_t)^\top \right) \left( \lambda_i I_d + \sum_{j \in \mathcal{I}_i} \sum_{t=1}^{T-1} (o_{j,t} - z_t)(o_{j,t} - z_t)^\top \right)^{-1}.$$

$$\tag{20}$$

*Proof.* The proof is similarly applied by looking at the zero gradient of $M_i$. $\qquad\square$

If we set $\lambda_i = 0$ and $z = 0$, the minimization reduces back to estimation of $\hat{M}$ in Eq. (5).

## A.3 ERROR ANALYSIS

**Lemma A.5.** *Let $M_i^*$ be the true dynamic of the underlying state, we have:*

$$z(\{M_i^*\}) - z^* = -\lambda_z (\Lambda_n^*)^{-1} z^* + (\Lambda_n^*)^{-1} \left( \sum_{i=1}^{k} \sum_{j \in \mathcal{I}_i} A_i^\top \varepsilon_j \right), \tag{21}$$

*where $\Lambda_n^* = \lambda_z I_{d \times T} + \sum_{i=1}^{k} n_i G_i^*$.*

*Proof.* By expand the definition of $z(\{M_i^*\}$, we have:

$$z(\{M_i^*\}) = (\Lambda_n^*)^{-1} \left( \sum_{i=1}^{k} \sum_{j \in \mathcal{I}_i} G_i o_j \right)$$

$$= (\Lambda_n^*)^{-1} \left( \sum_{i=1}^{k} \sum_{j \in \mathcal{I}_i} A_i^\top (A_i o_j) \right)$$

$$= (\Lambda_n^*)^{-1} \left( \sum_{i=1}^{k} \sum_{j \in \mathcal{I}_i} A_i^\top (A_i z^* + \varepsilon_j) \right)$$

$$= (\Lambda_n^*)^{-1} \left( \Lambda_n^* z^* - \lambda_z z^* + \sum_{i=1}^{k} \sum_{j \in \mathcal{I}_i} A_i^\top \varepsilon_j \right)$$

$$= z^* - \lambda_z (\Lambda_n^*)^{-1} z^* + (\Lambda_n^*)^{-1} \left( \sum_{i=1}^{k} \sum_{j \in \mathcal{I}_i} A_i^\top \varepsilon_j \right)$$

$\qquad\square$

**Lemma A.6.** *Let $z^*$ be the true exogenous noise, we have:*

$$M_i(z^*) - M_i^* = \left( \lambda_i(I_d - M_i^*) + \sum_{j \in \mathcal{I}_i} \sum_{t=1}^{T-1} \varepsilon_{j,t}(o_{j,t} - z_t^*)^\top \right) (\lambda_i I_d + \Sigma_n^*)^{-1}, \qquad (22)$$

*where $\Sigma_n^* = \sum_{j \in \mathcal{I}_i} \sum_{t=1}^{T-1} (o_{j,t} - z_t^*)(o_{j,t} - z_t^*)^\top$ is the empirical covariace matrix.*

*Proof.* By expand the definition of $M_i(z^*)$, we have:

$$M_i(z^*) = \left( \lambda_i I_d + \sum_{j \in \mathcal{I}_i} \sum_{t=1}^{T-1} (o_{j,t+1} - z_{t+1}^*)(o_{j,t} - z_t^*)^\top \right) \left( \lambda_i I_d + \sum_{j \in \mathcal{I}_i} \sum_{t=1}^{T-1} (o_{j,t} - z_t^*)(o_{j,t} - z_t^*)^\top \right)^{-1}$$
$$(23)$$

$$= \left( \lambda_i I_d + \sum_{j \in \mathcal{I}_i} \sum_{t=1}^{T-1} (\varepsilon_{j,t} + M_i^*(o_{j,t} - z_t^*)) (o_{j,t} - z_t^*)^\top \right) (\lambda_i I_d + \Sigma_n^*)^{-1} \qquad (24)$$

$$= M_i^* + \left( \lambda_i(I_d - M_i^*) + \sum_{j \in \mathcal{I}_i} \sum_{t=1}^{T-1} \varepsilon_{j,t}(o_{j,t} - z_t^*)^\top \right) (\lambda_i I_d + \Sigma_n^*)^{-1} \qquad (25)$$

$\square$

### A.4 PROOF OF PROPOSITION 2.5

*Proof.* By induction, it is not hard to prove that

$$\mathbb{E}[O_t | O_0 = o, D = i] = M_i^t o.$$

Sum up all condition on $O_0$, we have:

$$\mathbb{E}[O_t | D = i] = M_i^t \mathbb{E}[O_0].$$

By the definition of long-term discounted reward $G$, we have:

$$\begin{aligned} v(\pi_i) &= \mathbb{E}[\sum_{t=0}^{\infty} \gamma^t R_t | D = i] \\ &= \sum_{t=0}^{\infty} \gamma^t \mathbb{E}[\theta_r^\top O_t | D = i] \\ &= \theta_r^\top \sum_{t=0}^{\infty} \gamma^t M_i^t \mathbb{E}[O_0] \\ &= \theta_r^\top (I - \gamma M_i)^{-1} \mathbb{E}[O_0], \end{aligned}$$

where the last equation holds when $\|M_i\| < \frac{1}{\gamma}$. $\square$

### A.5 PROOF OF PROPOSITION 3.5

*Proof.* From Lemma A.5, suppose $\lambda_z = 0$ and $(\Lambda_n^*)^{-1}$ exists, we have:

$$z - z^* = (\Lambda_n^*)^{-1} \left( \sum_{i=0}^{1} \sum_{j \in \mathcal{I}_i} A_i^\top \varepsilon_j \right).$$

Consider $\hat{v}(\pi_0)$ if we plugin $\hat{z}$ and the true dynamic $M_0^*$, the error between $\hat{v}$ and $v$ is

$$\begin{aligned} \hat{v}(\pi_0) - v(\pi_0) &= \theta_r^\top (I - \gamma M_0^*)^{-1}(z_0 - z_0^*) \\ &:= \beta_r^\top (z_0 - z_0^*) \\ &= (\beta_r^\top, 0, \ldots, 0)(z_0 - z_0^*) \\ &= \tilde{\beta}_r^\top (z_0 - z_0^*), \end{aligned}$$

where $\beta_r = (I - \gamma M_0^*)^{-T}\theta_r$, and $\tilde{\beta}_r$ is the extended vector of $\beta_r$ if we fill the other vector value at other time step as $0$.

Expand the difference $(z_0 - z_0^*)$ we have:

$$
\begin{aligned}
\hat{v}(\pi_0) - v(\pi_0) =& \tilde{\beta}_r^\top (z_0 - z_0^*) \\
=& \sum_{i=0}^{1} \tilde{\beta}_r^\top (\Lambda_n^*)^{-1} A_i^\top \big(\sum_{j\in\mathcal{I}_i} \varepsilon_j\big) \\
=& \sum_{i=0}^{1} \tilde{\beta}_r^\top (\frac{\Lambda_n^*}{n})^{-1} A_i^\top \big(\frac{\sum_{j\in\mathcal{I}_i} \varepsilon_j}{n}\big) \\
\leq& \|\tilde{\beta}_r\| \sum_{i=0}^{1} \|(\frac{\Lambda_n^*}{n})^{-1} A_i\| \|\frac{\sum_{j\in\mathcal{I}_i} \varepsilon_j}{n}\|.
\end{aligned}
$$

By Assumption A.1 and Assumption A.2, the norm of $\tilde{\beta}_r$ is the same as $\beta_r$, which is bounded by $\|\beta_r\| \leq \frac{1}{1-\gamma C_{M_i}}\|\theta_r\|$. The matrix norm in the middle factor is bounded because of Assumption A.2. Finally, by vector concentration inequality, since $\varepsilon_j$ is norm-subGaussian (Jin et al., 2019), there exist a constant $c$ that with probability at least $1 - \delta$:

$$
\|\frac{\sum_{j\in\mathcal{I}_i} \varepsilon_j}{n}\| \leq c\sqrt{\frac{\log(2dT/\delta)}{n}}.
$$

In sum, the error is bounded by $\mathcal{O}(\frac{1}{\sqrt{n}})$ with probability at least $1 - \delta$, and the constant depends on $C_\varepsilon, C_{M_i}, C_\Lambda$ and the norm of $\|\theta_r\|$. $\qquad\square$

## A.6 PROOF OF PROPOSITION 3.6

*Proof.* Since we get access to the ground true $z^*$, the remaining problem is by changing the state as $s_{j,t} = o_{j,t} - z_t^*$ and reduce the problem back to standard MDP. The detailed proof can refer to Proposition 11 in Miyaguchi (2021). $\qquad\square$

# B    REDUCE THE COMPUTATION COMPLEXITY WITH PRE-COMPUTATION

In this section, we explain how to reduce the computation complexity with pre-computation.

**Pre-computation.**    Compute

$$M_i(0) = \left(\sum_{j \in \mathcal{I}_i} \sum_{t=1}^{T-1} o_{j,t+1} o_{j,t}^\top\right) \left(\sum_{j \in \mathcal{I}_i} \sum_{t=1}^{T-1} o_{j,t} o_{j,t}^\top\right)^{-1}$$

and

$$\bar{o}_t = \sum_{j \in \mathcal{I}_i} o_{j,t}.$$

The pre-computation requires computation complexity of $\mathcal{O}(nTd^2 + d^3)$, where $d^2$ is the computation complexity of the outer product, $d^3$ is the computation complexity of the matrix inversion after summing up the matrix.

**In Each Iteration.**    The computation of $M$ can be rewritten as

$$M_i(z) = \left(\sum_{j \in \mathcal{I}_i} \sum_{t=1}^{T-1} (o_{j,t+1} - z_{t+1})(o_{j,t} - z_t)^\top\right) \left(\sum_{j \in \mathcal{I}_i} \sum_{t=1}^{T-1} (o_{j,t} - z_t)(o_{j,t} - z_t)^\top\right)^{-1}$$

$$= M_i(0) - \sum_{t=1}^{T-1} z_{t+1} \bar{o}_t^\top - \sum_{t=1}^{T-1} \bar{o}_{t+1} z_t^\top + \sum_{t=1}^{T-1} z_{t+1} z_t^\top,$$

which requires computation complexity of $O(Td^2)$. Similarly, the computation of

$$z(G) = (\sum_{i=0}^{k} n_i G_i)^{-1} (\sum_{i=0}^{k} G_i \bar{o}_j)$$

requires computation complexity of $\mathcal{O}(T^2 d^2)$. Both steps are computationally scalable, since it does not rely on number of individuals $n$ (which is often much larger than $T$ and $d$).

**Overall Computation Complexity.**    Suppose we execute the iterations for $k$ times, then the total computation complexity for the alternating minimization is $\mathcal{O}(nTd^2 + d^3 + kT^2 d^2)$. In practice, the number of different individual $n$ is far larger than the experiment horizon $T$ and the feature dimension $d$, therefore the computation complexity essentially scales linearly with $n$.

# C  EXPERIMENTS DETAILS

## C.1  SYNTHETIC SIMULATION

The synthetic environment generates 4 randomized matrix $M_i$ for policies $\{\pi_i\}_{i=0}^3$, where each entry of $M_i$ is a positive number randomly sample from a uniform distribution between $(0, 1)$. We normalize each row so that it sums up to 1, and we set $\tilde{M}_i = 0.5I + 0.5M_i$ as our final transition matrix. The $0.5I$ part ensures each matrix is not too far away from each other.

We generate a set of i.i.d. random vector $\eta_t \sim \mathcal{N}(0, 1.5I)$ and set $z_{t+1} = z_t + \eta_t$ recursively. And we let $\tilde{z}_t = \alpha_t * z_t$ as the final exogenous noise, where $\alpha_t = e^{\beta_t}$ and $\beta_t \sim \mathcal{N}(0, 0.5I), i.i.d..$

All the parameters ($z_t$ and $M_i$) of the dynamic are fixed once generated, and we use the dynamic to generate our observation for each individual, following

$$s_{t+1} = M_i s_t + \varepsilon_t, \text{ and } o_t = s_t + \alpha z_t, \forall t$$

where $\varepsilon_t$ is independently drawn from a standard normal distribution, and $\alpha$ control the level of exogenous noise.

## C.2  POLICY CONSTRUCTION IN TYPE-1 DIABETE SIMULATOR

The Bagal and Bolus policy is a parametrized policy based on the amount of insulin that a person with diabetes is instructed to inject prior to eating a meal (Bastani, 2014)

$$\text{injection} = \frac{\text{current blood glucose} - \text{target blood glucose}}{\text{CF}} + \frac{\text{meal size}}{\text{CR}},$$

where $CF$ and $CR$ are parameter based on patients information such as body weights, which is already specified in the simulator.

We set our two treatment policies with target blood glucose level at $145$ and $130$ (compared to control: $140$). And we increase the noise in the insulin pump simulator in both the treatment policies.

## C.3  RANDOM PATIENTS GENERATION IN TYPE-1 DIABETE SIMULATOR

Type-1 Diabete simulator pre-stores 30 patients parameter. To randomly generate a new patient, we randomly pick two different patients $A$ and $B$, and use a random linear coefficient $\beta \sim U(0, 0.2)$ and mixed the parameter of a new patient as

$$\theta = (1 - \alpha)\theta_A + \alpha\theta_B,$$

where $\theta_A$ and $\theta_B$ are the parameters of patients A and B, respectively. Since patient $A$ has more weight of the parameter, the parameters in Bagal and Bolus policy, $CF$ and $CR$, follow patient $A$'s parameter.

## C.4  FULL RESULTS FOR ALL THE ONLINE STORE EXPERIMENTS.

|  | Metric 1 | Metric 2 | Metric 3 | Metric 4 |
|---|---|---|---|---|
| Naive Average | 122.47% | 93.61% | 51.20% | **25.28%** |
| Stationary | 174.77% | 454.87% | 61.71% | 110.58% |
| Non-stationary | **94.56%** | **12.54%** | **26.79%** | 67.57% |

Table 2: Experiment # 1

| | | Metric 1 | Metric 2 | Metric 3 | Metric 4 |
|---|---|---|---|---|---|
| Treatment 1 | Naive Average | **41.91**% | 66.09% | 3139.91% | 431.44% |
| | Stationary | 51.60% | **48.03**% | 4471.33% | 275.49% |
| | Non-stationary | 64.68% | **48.28**% | **770.27**% | **13.59**% |
| Treatment 2 | Naive Average | 236.13% | 43.44% | 106.85% | 122.65% |
| | Stationary | 150.97% | **12.07**% | 98.31% | 199.95% |
| | Non-stationary | **10.69**% | 62.00% | **48.04**% | **109.96**% |

Table 3: Experiment # 2

| | Metric 1 | Metric 2 | Metric 3 | Metric 4 |
|---|---|---|---|---|
| Naive Average | 396.54% | **79.88**% | 192.84% | 17.21% |
| Stationary | 697.59% | 123.43% | 364.49% | **6.73**% |
| Non-stationary | **98.37**% | 81.86% | **30.65**% | 12.89% |

Table 4: Experiment # 3

| | | Metric 1 | Metric 2 | Metric 3 | Metric 4 |
|---|---|---|---|---|---|
| Treatment 1 | Naive Average | 8078.75% | 43.18% | 208.55% | 2438.43% |
| | Stationary | 7386.81% | **7.20**% | 154.66% | 1889.47% |
| | Non-stationary | **2328.03**% | 114.34% | **25.18**% | **102.84**% |
| Treatment 2 | Naive Average | 126.70% | 138.98% | 38.60% | **37.99**% |
| | Stationary | 172.57% | **12.01**% | **10.44**% | 46.60% |
| | Non-stationary | **29.92**% | 72.62% | 54.16% | 69.75% |
| Treatment 3 | Naive Average | 133.61% | 45.67% | 50.87% | 17.01% |
| | Stationary | 258.88% | 88.77% | **27.11**% | **12.89**% |
| | Non-stationary | **24.58**% | **34.89**% | 74.14% | 66.32% |

Table 5: Experiment # 4

