# OpenReview forum: "A Reinforcement Learning Approach to Estimating Long-term Treatment Effects"
_ICLR.cc/2023/Conference — Submitted to ICLR 2023_

### Official Review · Reviewer_67md · 2022-10-22

**Confidence:** 4
**Correctness:** 4
**Technical Novelty And Significance:** 2
**Empirical Novelty And Significance:** 2
**Recommendation:** 6

**Clarity, Quality, Novelty And Reproducibility:**

The paper is well written, and all points are clearly presented.  The extension over the traditional RL work is not substantial, as we all know one can relax some model assumptions to better serve a real-world problem. The question is how much more data and computation is required to justify the benefits.  This is not clearly articulated by the authors.

**Strength And Weaknesses:**

Strengths:
1. The problem statement is well written.
2. Introducing complexity into RL to deal with a real-world problem is good.

Weaknesses:
1. This work can be regarded as an intermediate work toward a milestone work.  The datasets and data utilized to validate effective’s cannot “prove” the proposed algorithm is “the algorithm” to address the issue.
2. Long term, but how long is long term?  How can the proposed method be used to predict financial market, for instance.  Would the linearity assumption be an issue for many real-world problems?

**Summary Of The Paper:**

This work proposes a non-stationary RL framework to make long-term predictions on treatment effects.  The proposed algorithm shows better results performed in two synthetic datasets and one online store dataset.

**Summary Of The Review:**

The paper deals with a real problem in medicine trials.  The proposed algorithm is theoretically sound.  The empirical results so show good effects.  The remaining question is that can the algorithm really make practical impact to clinical trials, or this is just an experimental method of many possibilities.  If the authors can specify how realistic to collect sufficient training data to learn model parameters of such high-complexity model and how realistic that the model variances can be tracked and modeled would be helpful.

---

> ### Author Response · Authors · 2022-11-11
> **Response to Reviewer 67md**
>
> Thank you for your positive comments and valuable feedbacks.
>
> [Q1. Motivating problem, Real world problem in this paper. Scope of the paper.] Our motivating problem is on the large scale e-commerce problem, especially on predicting the long-term effect prediction in an online store. Type-1 Diabetes is merely a simulation environment used as a benchmark to compare our algorithm with baselines. The real world data (Section 5.3) is an online store dataset. We disagree respectfully that our work is “intermediate”. Instead, our work provides a full solution for a challenging problem, and is among the few that includes validation on large-scale real-world commercial datasets.
>
> [Q2. How long can the proposed method be used to predict financial market?]  One can use the method to predict multi-month or even year-long impacts. Our online store experiment predicts 12-week long-term prediction from 5 weeks experiment (Section 5.3) only because we want to use available data for validation purposes.
>
> [Q3. Linear assumption] Please see general response Q1.

---

### Official Review · Reviewer_6c7X · 2022-10-24

**Confidence:** 4
**Clarity, Quality, Novelty And Reproducibility:** 1. The definitions of the important q…
**Correctness:** 2
**Technical Novelty And Significance:** 2
**Empirical Novelty And Significance:** 2
**Recommendation:** 3

**Strength And Weaknesses:**

Strength: The paper proposes a treatment effect estimator in non-stationary dynamics. The authors justify the estimator both theoretically and empirically.

Weaknesses:
1. The authors claim to estimate the average long-term rewards. However, the definition in equation (1) is different from the discounted sums of the rewards. From this perspective, the long-term average reward in this paper is
not the typical setting in RL literature \citep{liao2020batch, liao2021off}. This might
 mislead the readers.

2. The paper makes linear assumptions, i.e., Assumptions 2.1 and  2.3, on transition kernel and reward. This assumption can be regarded as the alignment of the linear MDP assumption in RL literature. However, in OPE method referred to in the paper, all of the visitation-based approaches do not require such linear assumptions.

3. Assumption 3.1. The linear decomposition of the observation $O_{t}$ is restrictive. In a real-world environment, endogenous and exogenous noise is hard to be differentiated by just following a linear way. Otherwise, the author should provide some empirical analysis on whether the decomposition is indeed following a linear decomposition.

4. Is the Monte Carlo sample estimator $\hat{S}_0$ having a large variance when there exists a large distribution shifting of the off-policy data? In this case, how to control the variance of the estimator?

5. The computational intensity analysis should be provided. The alternating optimization algorithm over $M_0$, $M_1$, and $z$ seem to lead to a non-trivial optimization problem. In addition, could the authors show that the algorithm is convergent in a heuristic sense?

6. Proposition 3.5. The finite sample bound should be explicitly provided not just the informal rate of convergence. In another sense, the Markov process is non-stationary and the data is dependent. I cannot find the parts the authors address such problems when deriving the theoretical results.

reference:
Liao, P., Klasnja, P., and Murphy, S. (2021), “Off-policy estimation of long-term aver-
age outcomes with applications to mobile health,” Journal of the American Statistical
Association, 116, 382–391.
Liao, P., Qi, Z., Klasnja, P., and Murphy, S. (2020), “Batch policy learning in average
reward markov decision processes,” arXiv preprint arXiv:2007.11771.



**Summary Of The Paper:**

The paper leverage the variant of the OPE estimator to estimate the average treatment effects in non-stationary dynamics. The problem is interesting and challenging, however, the method proposed in the paper might be limited for practical use.


**Summary Of The Review:**

In summary, solving estimating treatment effects in non-stationary dynamics is challenging and interesting. However, this work does not provide a rigorous step for solving the problem and requires some additional assumptions.

---

> ### Author Response · Authors · 2022-11-11
> **Response to Reviewer 6c7X**
>
> Thank you for your detailed comments and valuable suggestions!
>
> [Q1. Definition of Eq.(1) and the connection to the long-term average reward] In our definition, the long-term treatment value is the *difference* between two policy value $\Delta = v(\pi_1) - v(\pi_0)$, where $v(\pi_i)$ is the expected long-term average reward of $\pi_i$, and can be estimated by our method.
>
> [Q2. Visitation-based method does not need linear assumption] See general response Q1.
>
> [Q3 Linear additive noise] The linear additive noise model enables efficient solutions, and is directly motivated by real world data. Figure 1 shows the metric difference between treatment and control policies is much more stable than metrics themselves. The intuition is clear: both treatment and control metrics are similarly affected by exogenous factors like seasonal effects. Therefore, we consider the additive noise model to be important and practically relevant.
>
> [Q4. Estimation of initial distribution] There is *no* distribution mismatch in initial state distribution between control and treatment, because at the beginning of an experiment individuals are randomly assigned. Therefore, the variance from such Monte Carlo estimates (by averaging observed initial states) is low.
>
> [Q5. Computation analysis and convergent analysis] For computation, we can pre-compute $\hat{M}$ without z in Eq. (13), as well as the average vector of the observation. The update of $\hat{M}$ with $z$ in Eq. (13) can be decomposed into pre-computing $\hat{M}$ and the involvement of $z$ outer-product the average observation in some way. Therefore, each alternating step (after pre-computation) has computation complexity of $O(d^3)$ with no dependency on the size of sample $n$, so the computation cost is low even for fairly large d. We will include the computation analysis in the revision version.
>
> For convergence, heuristically speaking, it converges because the loss function always decreases when conducting alternating minimization, thus it will converge eventually. In practice, we use 20 alternating steps and both z and M stabilize around step 5.
>
> [Q6. Finite sample bound and data dependency] See general response Q3. Our theoretical analysis allows data to be generated by a Markov process, thus samples can be dependent. The analysis is based on a martingale structure in Eq. (16), once the residual is bounded and expected to be 0 given the filtration on previous info.
>
> [Q7. Definition of $R_t, π_0$ and $\hat{Δ}$]
>
> * $R_t$ and reward function $r$: thanks for pointing it out. We accidentally commented out the definition in the submission, and have fixed it.
> * $\pi_0$: control policy, defined in the first paragraph in Section 2.1.
> * $\hat{\Delta}$: Estimation of the long-term treatment effect, defined at the end of Section 2.1.
>
>
> [Q8. Clarify Contribution of Equation (5) (stationary part)] We do not claim it to be our contribution. Instead, the sentence right after Equation (5) cites (Parr et al., 2008) for a detailed derivation.
>
> [Q9. What's the estimator $\hat{z}$ representing?] $z$ is the exogenous noise outside of the control in the environment, in the online-store example, the magnitude of noise can be interpreted as variations (e.g. seasonal effects) and trends (e.g. economic environment) in the whole market place.
>
> [Q10. Related work on causal inference] See general response Q2.

---

> > ### Comment · Reviewer_6c7X · 2022-12-04
> > **Response to Authors**
> >
> > Thank you for the detailed responses and address my comments, the paper's quality is relative improved in the current version. However, there are still three main issues for the paper:
> >
> > 1. The linear additive noise structure is still limited in practical use [1], which limits the potential impact and contribution of the work to the community.
> >
> > 2. The empirical analysis is not adequate, there are many comparable works are not implemented for numerical comparison in the paper.
> >
> > 3. The algorithm and the implementation details are not fully disclosed, I mean the code is not shared which limits the credibility of the work. Although the authors justify the potential computational issue of the work, however it is still not convincing without support of the code implementation.
> >
> > So I would like to keep my score.
> >
> > [1] Tchetgen, E. J. T., Ying, A., Cui, Y., Shi, X., & Miao, W. (2020). An introduction to proximal causal learning. arXiv preprint arXiv:2009.10982.

---

> > > ### Author Response · Authors · 2022-12-07
> > > **Response to Reviewer 6c7X**
> > >
> > > Thank you for the comment. Please see our point-to-point responses:
> > >
> > > 1. The reference paper studies the identification of the potential outcome under unobserved confounders. However, although the paper points out that "The system of linear structural equations considered above is overly restrictive", the setting of proximal causal inference differs from ours, where it requires additional assumption on the causal structure and the observation of two proxy variable Z and W. In addition, it does not explicitly aim for estimating long-term effects. Given the irrelevancy of the two papers, we do not see how the reference leads to the conclusion that linear models are overly restrictive. In fact, many large-scale applied works in the industry are based on linear models due to their simplicity, fast inference time, and interpretability.
> > > 2. Could you specify a list of algorithms that we need to compare with?
> > > 3. Although the code is not provided (due to our internal policy), the computation complexity analysis in the appendix demonstrates scalability of our approach. This is the standard way to quantify an algorithm’s worst-case complexity in computer science. Could you specify which part of the analysis is not convincing and can lead to potential computation issue?

---

### Official Review · Reviewer_XjcA · 2022-10-24

**Confidence:** 4
**Correctness:** 3
**Technical Novelty And Significance:** 3
**Empirical Novelty And Significance:** 3
**Recommendation:** 5

**Clarity, Quality, Novelty And Reproducibility:**

The presentation is clear in general.

The quality is good. But the paper would benefit from a substantial revision to better highlight the contribution, relax the linearity assumption, justify the use of reinforcement learning, formulate the problem based on hypothesis testing, provide rigorous uncertainty quantification, conduct more detailed theoretical analysis, use a different simulation setting and include the link for the code.

The main novelty includes the development of a practical algorithm for long-term treatment effect evaluation under nonstationarity. The associated theoretical analysis is novel as well. Nonetheless, some of the claimed contributions have been developed and employed in the existing literature.

**Strength And Weaknesses:**

Strengths:

1. Nonstationarity is commonly seen in real-world applications. Most existing works on policy evaluation did not take nonstationarity into consideration. The paper takes the issue of nonstationarity seriously and borrows ideas from the economics literature to deal with nonstationary environments.

2. A practical algorithm is developed for long-term treatment effect evaluation under nonstationarity. The algorithm is also easier to implement. Various practical considerations are discussed and several extensions are outlined.

3. Theoretical justifications of the proposed algorithm are provided. In addition, one online-store dataset is also employed to evaluate the proposed algorithm in real applications.

Weaknesses:

1. Missing literature on A/B testing & causal inference. There is a huge literature on A/B testing. In addition, there is a growing literature on estimating long term treatment effects in causal inference, see e.g., https://scholar.harvard.edu/files/shephard/files/cause20170718.pdf and the papers that cited this paper. These works are not discussed in the paper, but shall be included and potentially contrasted (see also point #4 below). More important, there are some prior works that proposed to use reinforcement learning for long-term treatment effects estimation in A/B testing, see e.g., https://www.tandfonline.com/doi/full/10.1080/01621459.2022.2027776. They also adopt ideas from the off-policy evaluation literature. The author(s) might want to discuss in detail the difference from these papers.

2. The contributions are somehow overstated given the prior work on applying reinforcement learning to long-term treatment effects estimation. I suggest the author(s) to focus on the issue of nonstationarity and revise the contribution, the introduction section and the summary. You might also want to include "nonstationarity/nonstationary environments" in the title to highlight the contributions of the paper more accurately.

3. The linear MDP assumption is strong and shall be relaxed if possible.

4. The use of reinforcement learning framework is not well-justified. In particular, under the current experimental design, each subject receives one static treatment all the time. Existing A/B testing methods are also applicable for causal effect estimation. The paper would benefit from a detailed discussion about the advantage of employing reinforcement learning over standard A/B testing methods.

5. Uncertainty quantification is not studied in the paper. In additional to the point estimator, in A/B testing, decision makers are equally interested in understanding if a new product is significantly better compared to an old one or not. The author(s) might want to formulate the problem using hypothesis testing and develop a rigorous testing procedure to test these hypotheses.

6. Some of the descriptions are not very accurate and some details are missing. For instance, in my opinion, off-policy evaluation might not be very related to the problem the author(s) studied. In particular, off-policy evaluation considers the scenario where the behavior policy differs from the target policy. However, in the current setting, each subject receives one of the two target policies all the time. This is essentially an "on-policy" (as apposed to off-policy) setting.

7. In Propositions 3.5 and 3.6, asymptotic rate of convergence is provided to quantify the difference between the proposed estimator and the ground truth. It would be better to develop nonasymptotic error bounds not only as a function of the sample size, but other relevant parameters in the problem as well.

8. The type-I diabetes setting is not very realistic. In practice, it would be impossible to get data for over 10 thousand patients. It might be better to use another environment if your method requires a large number of trajectories.

9. I might miss something, but I did not find the link for the code. So cannot check the reproducibility of the numerical experiments.


**Summary Of The Paper:**

The paper adopts a reinforcement learning framework to estimate the long-term treatment effects in nonstationary environments. The main contributions lies in the development of a practical algorithm to estimate causal effect under nonstationarity. The algorithm is justified via both theoretical results, synthetic environments and a real-world online store dataset.

**Summary Of The Review:**

As I mentioned earlier, the paper contains some interesting ideas on dealing with nonstationarity. Nonetheless, it needs to be heavily revised to better highlight the contribution, relax the linearity assumption, justify the use of reinforcement learning, formulate the problem based on hypothesis testing, provide rigorous uncertainty quantification and conduct more detailed theoretical and numerical analysis. However, it has the potential to become a high-impact paper if all the aforementioned weaknesses can be overcomed. I am very keen to increase my score if shall these comments be addressed.

---

> ### Author Response · Authors · 2022-11-11
> **Response to Reviewer XjcA**
>
> Thank you for your detailed comments and insightful suggestion!
>
> [Q1. Missing Literature] Please see general response Q2.
>
> [Q2. overstate the contribution and suggest revise the title] Thanks for the suggestion. We did not intend to claim our work is the first RL approach, and will make it clearer. Further, we will change the title to “A Reinforcement Learning Approach to Estimating Long-term Effects in Nonstationary Environments”, to emphasize the focus and contributions of this work.
>
> [Q3. Linear MDP assumption] Please see general response Q1.
>
> [Q4. The use of RL framework is not well-justified.] We want to clarify that standard A/B testing methods are insufficient in our setting. In standard A/B testing, we observe the outcome of a unit after it’s exposed to the treatment or control. In our setting, the outcome is the long-term value defined in Eq. (1). Applying standard A/B testing means running the experiment for a very long time, which is inefficient and expensive. The key challenge here is to predict such a *long-term* outcome based on *short-term* experiment data (which Section 3 is dedicated to). Please see the 2nd paragraph in Section 1 for more discussion.
>
> In the experiments (Section 5), we compared our method to Naive Averaging (a version of standard A/B testing for estimating short-term causal effects). Our results demonstrate the advantage of our method.
>
> [Q5. Uncertainty Quantification (UQ) and non-asymptotic error bound] See general response Q3.
>
> [Q6. Relation to OPE] The reviewer is right that our problem is not OPE, but our method is inspired by recent OPE advances (3rd paragraph in Introduction). That said, our setting does share certain similarities to off-policy evaluation: our goal is to estimate long-term impacts based on short-term data. The distribution of states in the long term differs from observed short-term distribution. Such a distribution shift between short-term data and long-term target is a shared technical challenges as OPE.
>
> [Q7. Type-I Diabetes setting is not realistic] We want to clarify that our focus is in developing a method for problems inspired by online stores, rather than advancing state of the art for medical problems like diabetes. The diabetes simulator is one of the benchmarks to test our algorithm, and we also use real-world data from an online store for empirical validation. While the simulator is not realistic (like many other medical simulators), it is useful for us to design and replicate experiments to gain insights for the proposed method. For example, it shows that even if the underlying dynamics are not linear, our method can still produce useful predictions. We will include more experiment details in the revision version in Appendix.
>
> [Q8. Code sharing] We do hope to share the code, but will have to go through a process due to proprietary nature of the code.

---

### Official Review · Reviewer_GJp8 · 2022-10-26

**Confidence:** 3
**Correctness:** 3
**Technical Novelty And Significance:** 2
**Empirical Novelty And Significance:** 3
**Recommendation:** 5

**Clarity, Quality, Novelty And Reproducibility:**

The overall presentation is good. It is not hard to understand the paper. The proposed algorithm and analysis are pretty natural. It seems that the code is not provided, so it is hard to judge the reproducibility.

**Strength And Weaknesses:**

The paper studies a practical and important problem: estimate long-term effect under nonstationary dynamics. The proposed algorithm is natural and simple.

My main concern is the linear assumptions. Is it possible to generalize the results for generalized linear models? Will the prediction value be pretty biased for generalized linear models? Another comment is that there are other papers that use reinforcement learning approach to estimate long-term effect, for example, [1] and literature on dynamic treatment regimes, and I think these papers need to be cited for comparison.


[1] Chengchun Shi, Xiaoyu Wang, Shikai Luo, Hongtu Zhu, Jieping Ye,and Rui Song, Dynamic Causal Effects Evaluation in A/B Testing with a Reinforcement Learning Framework, 2020.

**Summary Of The Paper:**

This paper proposes a reinforcement learning based algorithm to estimate long-term effect for a class of nonstationary problems. Empirical results in both synthetic and real datasets show the potential of the proposed algorithm.

**Summary Of The Review:**

As mentioned above, this paper attempts to tackle an important practical problem. Though the theoretical and empirical results justify the potential of the proposed algorithm, I think the linear assumptions are strong. If possible, I hope the authors could say or show something on generalized linear models and I would be happy to increase my score.

---

> ### Author Response · Authors · 2022-11-11
> **Response to Reviewer GJp8**
>
> Thank you for your positive feedback and valuable suggestion.
>
> [Q1. Generalized linear assumption] Please see general response Q1.
>
> [Q2. Citation] Please see general response Q2.

---

### Author Response · Authors · 2022-11-11
**General Response (part 1)**

We thank all reviewers for the detailed and valuable comments. We first address three common questions.

[Q1. Linear assumption and its limitation] We agree that the linear assumption has limitations. However, our method does not rely on the assumption, but can be interpreted as an optimization procedure that leverages linear representations. In our experiments of Type-1 Diabetes and Online-store, both environments dynamics are non-linear, but our linear representation can still produce more accurate predictions than the baselines. On the other hand, linear representation is popular in the RL literature (e.g. related work in [1]), and often preferable in industrial applications due to simplicity and greater model interpretability.

Our method can be extended to nonlinear function classes. Under the linear assumption,  our model-based method is equivalent to value-based and even visitation-based methods (see e.g. Theorem 1 in [2] for the detail derivation). Thanks to the equivalency to the linear value-based method, we can extend the linear approximator of the value function to any non-linear function class, e.g. Generalized Linear model or Neural Network, where  Therefore, one way to extend to nonlinear cases is to use Generalized Linear Models or neural networks to replace the linear value function, and modify our loss function (Eq. (9)) as Bellman or Temporal Difference errors. For example, if we use value-based approach, we can model our Q-function with a deep neural network, and use a Fitted-Q Evaluation (e.g. [3]) style to update the model parameter $$\theta \gets \arg\min_\theta \sum (q_\theta(o_t-z_t) - r_t - \gamma q_{\theta’}(o_{t+1}-z_{t+1}))^2,$$ where $\theta$ is the parameter of the Neural Network, and $\theta'$ is the parameter of the target network.

[Q2. Addition literature.] We thank reviewers for pointing out further literature, and will add a discussion of them to better contextualize our contributions. We highlight two of them below: standard time-series causal estimations (e.g. [4]) and RL-based approach by Shi et al. [1].

The focus of standard time-series causal estimations (e.g. [4]) is defining several estimands for experiments on single time series, which differs from ours in several ways. First, we only expose individuals for a short-term window. Secondly, our long-term effect is one special case of the estimands in time series, which is the difference between exposing to treatment all the time and exposing to control all the time. Thirdly, we don’t have probabilistic treatment in experiment, we only randomly separate individuals to different group and expose them to a constant (treatment/control) policy.

Shi et al. [1] is the first paper to introduce RL in long-term treatment effect, and their setting is similar to our stationary setting in Section 2.2 and they allow randomized/probablistic treatment in the experiment. In comparison, we move one step forward by extending the stationary Markov assumption to non-stationary, which better captures the real world data.

---

### Author Response · Authors · 2022-11-11
**General Response (part 2)**

[Q3. Uncertainty Quantification] We agree both uncertainty quantification and non-asymptotic error bound are important. However, our main contributions are in formulating a tractable class of nonstationary environments based on data in an online store, develop an algorithm and validate it on benchmarks. The theoretical derivation of UQ is outside of the scope of our paper and we will leave it as future work.

On the other hand, we are aware of several promising directions for UQ, and will discuss them in the revision. Since our method is inspired by OPE, once we get an estimate of the exogenous noise, we can extend our work to several recent high confidence interval estimation in OPE literature, such as bootstrapping (e.g. [5]), CoinDice [6], non-asymptotic CI (e.g. [7]) or standard Central Limit Theorem like [1].

[1] Shi, Chengchun, Xiaoyu Wang, Shikai Luo, Hongtu Zhu, Jieping Ye, and Rui Song. "Dynamic causal effects evaluation in A/B testing with a reinforcement learning framework." Journal of the American Statistical Association (2022): 1-13.
[2] Yaqi Duan, Zeyu Jia, and Mengdi Wang. Minimax-optimal off-policy evaluation with linear function approximation. In International Conference on Machine Learning, pp. 2701–2709. PMLR, 2020.
[3] Voloshin, Cameron, Hoang M. Le, Nan Jiang, and Yisong Yue. "Empirical study of off-policy policy evaluation for reinforcement learning." arXiv preprint arXiv:1911.06854 (2019).
[4] Bojinov, Iavor, and Neil Shephard. "Time series experiments and causal estimands: exact randomization tests and trading." Journal of the American Statistical Association 114, no. 528 (2019): 1665-1682.
[5] Kostrikov, Ilya, and Ofir Nachum. "Statistical bootstrapping for uncertainty estimation in off-policy evaluation." arXiv preprint arXiv:2007.13609 (2020).
[6] Dai, Bo, Ofir Nachum, Yinlam Chow, Lihong Li, Csaba Szepesvári, and Dale Schuurmans. "Coindice: Off-policy confidence interval estimation." Advances in neural information processing systems 33 (2020): 9398-9411.
[7] Feng, Yihao, and Ziyang Tang. "Non-asymptotic Confidence Intervals of Off-policy Evaluation: Primal and Dual Bounds." In International Conference on Learning Representations. 2020.

---

### Author Response · Authors · 2022-11-18
**Revision on the paper**

Thank you again for the valuable suggestion on the paper revision, we have uploaded our revised paper. We highlight the modified content in blue color, and summarize as follow:

* We change the title.
* We add back the definition of $R_t$ and reward function in Section 2.1.
* We add a sentence to discuss the reason we use linear assumption in Section 2.2.
* We add discussion on several related works in Section 4, and we highlight Shi et al. in the introduction to better contextualizing our contribution.
* We add computation analysis in Appendix B.
* We add a detail experimental set up on how to choose random patients in Appendix C.4.

---

### Decision · Program_Chairs · 2023-01-20

**Decision:**

Reject

**Justification For Why Not Higher Score:**

Much other related work in the area of long-term effects (including in plain policy evaluation, even if no policy optimization is present) could be more critically appraised in the context of the novel methods being introduced - in particular, much of what is said in terms of advantages of RL compared to A/B testing seems to require stronger elaboration ("finding a reliable proxy is challenging in practice" is a blanket statement that would benefit from more refinement); it feels a bit modest in scope, by e.g. not attempting a fuller treatment of uncertainty quantification, so important for this problem.

**Justification For Why Not Lower Score:**

N/A

**Metareview: Summary, Strengths And Weaknesses:**

Estimating long-term effects is not an easy task, and in many cases involves high-stakes decision problems. This paper proposes methods for this problem in the context of reinforcement learning, with the extra challenge of non-stationary dynamics.

Strengths: a clear presentation; real-data experiments; an emphasis on easy-to-explain methods, which is always a plus (again, due to the nature of some application in this domain)

Weaknesses: much other related work in the area of long-term effects (including in plain policy evaluation, even if no policy optimization is present) could be more critically appraised in the context of the novel methods being introduced - in particular, much of what is said in terms of advantages of RL compared to A/B testing seems to require stronger elaboration ("finding a reliable proxy is challenging in practice" is a blanket statement that would benefit from more refinement); it feels a bit modest in scope, by e.g. not attempting a fuller treatment of uncertainty quantification, so important for this problem.